# Understanding HPV Vaccination Policymaking in Rwanda: A Case of Health Prioritization and Public-Private-Partnership in a Low-Resource Setting

**DOI:** 10.3390/ijerph20216998

**Published:** 2023-10-30

**Authors:** Eric Asempah, Mary E. Wiktorowicz

**Affiliations:** School of Health Policy & Management, Faculty of Health, York University, 4700 Keele Street, Toronto, ON M3J 1P3, Canada; mwiktor@yorku.ca

**Keywords:** Rwanda, policymaking process, vaccination, cervical cancer, HPV, health policy

## Abstract

Rwanda is the first African country to implement a national HPV vaccination program in 2011. This study sought to clarify the HPV vaccination policymaking process in Rwanda through the lens of Kingdon’s multiple stream framework and Foucault’s concept of governmentality. Perspectives of policymakers engaged in HPV vaccination policy were gathered from published sources, along with key informant interviews. Rwanda’s track record of successful vaccination programs enabled by a culture of local accountability created public and private sector incentives. Effective stakeholder engagement, health priority setting, and resource mobilization garnered locally and through international development aid, reflect indicators of policy success. The national HPV policymaking process in Rwanda unfolded in a relatively cohesive and stable policy network. Although peripheral stakeholder resistance and a constrained national budget can present a threat to policy survival, the study shows that such factors as the engagement of policy entrepreneurs within a policy network, private sector incentives, and international aid were effective in ensuring policy resolution.

## 1. Introduction

Most Rwandans reside in rural agricultural settings with limited access to healthcare [1,2]. The Rwandan genocide left the country’s healthcare system in complete shambles. By the end of the war in 1994 and the nation’s rebuilding process in the aftermath, the government made health a key priority. In 1998, the government launched a national development plan (often referred to as Vision 2020) that aimed to make Rwanda a middle-income nation by 2020 [2]. As a strategic step, a mutual health insurance scheme (also known as mutuelles de santé or mutuelles) was initiated in 1999, which ensured every citizen had some form of health insurance [3,4,5].

Twenty years later, Rwanda has made significant improvements in its health sector, increasing total health expenditure (THE) per capita from US$17 in 2003 to US$34 in 2006 [6]. In 2002, the Rwandan government allocated 8.6% of government revenue to health, that rose to 11.5% by 2010 [7]. The government allocated 200.8 billion Rwandan francs (approximately 200.8 million USD) to the health sector in 2018/19, an increase of 1.8% from the 2017/18 budgetary allocation of 197.4 billion Rwandan francs (approx. 197.4 million USD) [8].

The mutual health insurance scheme has been a pillar of the country’s framework for attaining Universal Health Coverage (UHC) [9,10,11] and includes over three-quarters of the population; the highest enrollment in health insurance in sub-Saharan Africa [9]. Rwanda’s health insurance program reportedly pivoted on three levels of public policymaking ideas; (1) problem definition, (2) practical ideas, and (3) policy ideas [9]. Rwanda has shown continuous dedication and commitment to disease prevention interventions, and consistently reports over 95% coverage in childhood vaccination [12,13].

Over the past decade, Rwanda has consistently incorporated ideals of health equity, value for money, and quality in its domestic policies [7,14]. The Rwandan government enshrined a commitment to prioritize health as a human right in its constitution (Article 41), stating that “All citizens have rights and duties relating to health”. The State is responsible for mobilizing activities aimed at promoting good health and assists in implementing them. In April 2011, the country initiated its nationwide HPV vaccination program with 93,888 (95.05% coverage) primary grade six girls receiving their first-dose of Gardasil^®^ at no cost to them [12,14,15,16]. The second and third dose for the same cohort recorded 89,704 (93.90% coverage) and 88,927 (93.23% coverage), respectively [14]. Between 2011 and 2018, 1,156,863 girls received their first dose of the HPV vaccine [17]. HPV vaccination for girls continues with a high rate coverage in Rwanda, while cervical cancer screening for women has also increased [18,19].

Multiple factors contributed to the success of Rwanda’s vaccination program according to Bao and colleagues, including “strong, high-level political will, multilevel accountability, effective use of funding, partnership with development partners, integrated health information, and community-level data collection” [20]. Rwanda’s resilience led it to overcome its past troubled history, crumbling healthcare before 1995, and economic setbacks, to become the first African nation to initiate a successful national HPV vaccination program with high coverage. On 30 May 2019, Sophie Cousins’s article “Why Rwanda could be the first country to wipe out cervical cancer” was captioned in CNN’s health column highlighting the government’s health prioritization towards eliminating cervical cancer [13]. As the first African nation with a nationwide HPV vaccination program, Rwanda sets a baseline for other African countries, particularly those that have yet to incorporate HPV vaccination into their national immunization program.

This paper seeks to elucidate the HPV vaccination policymaking process in Rwanda and identify the lessons learned to inform policymaking in other low-resource settings., Using Kingdon’s Multiple Stream Framework as the theoretical lens sheds lights on the policymaking process by clarifying the governance model, actions of policy entrepreneurs in public-private partnerships and the enduring interactions among them. The study is informed by data from primary and secondary sources.

## 2. Methods

Research ethics approval (certificate #STU 2021-137) was attained from University’s Ethics Review Board, Office of Research Ethics (ORE). Perspectives of policy makers engaged in HPV vaccination policy were gathered from published sources and key informant interviews. Key informants with an interest in cervical cancer prevention and control in Rwanda were identified using Google search. A list of potential stakeholders was assembled and emailed to seek their participation. The inclusion criteria were policymakers, politicians, opinion leaders, women’s advocacy groups that focus on women’s health, and academicians in Rwanda with research interest in cervical cancer. Stakeholders whose interest did not focus on cervical cancer prevention and control in Rwanda were excluded. One key informant (R001), a senior executive of a women’s advocacy and empowerment in Kigali, was interviewed via Zoom. A signed consent form was received prior to the interview via email, together with completed additional open-ended questionnaires on policymaking in Rwanda. A second key informant, an international policy actor, instrumental in the HPV vaccination policymaking in Rwanda attending the 35th International Papillomavirus Conference held on 17–21 April 2023, in Washington DC, USA provided information through personal communication with verbal consent. Most key informants initially identified were unable to participate due to competing commitments, while some did not respond to the request for an interview. Secondary data were obtained using relevant academic and grey literature, Rwandan government documents, and online newsletters to inform the Rwandan HPV nationwide vaccination policymaking process.

Databases searched included PubMed, Scopus, and Google Scholar. Only articles that provided information on the HPV vaccination program in Rwanda were assessed for data on the policymaking process. Google search was used to find grey literature, policy documents published by the Rwandan government, and online news items pertaining to the Rwandan HPV vaccination program.

### Theoretical Lens

Kingdon’s Multiple Stream Framework (MSF) and Foucault’s concept of governmentality were used to clarify the Rwandan policymaking process concerning the introduction of the national HPV vaccination program in 2011. Kingdon’s MSF involves consideration of three streams (problem, politics, and policy) that dynamically “interact to produce windows of opportunity” for action during governmental agenda setting [21]. When all streams converge, a window of opportunity opens to address the policy issue. Kingdon’s MSF focuses on how the development of agenda-setting ideas align with the problem, political and policy streams at the right moment to garner the needed attention to foster policy change. The propelling force for these ideas could be policy entrepreneurs who advocate for a particular position, interest, or goal in return for future benefits of the policy position advocated. According to Roberts and King, policy entrepreneurs are public entrepreneurs who, from outside the formal roles of government, introduce, translate, and help implement new ideas [22]. They serve as essential policy gap closers in the policy process. Policy entrepreneurs build relationships and tactically relay the problem(s) that need to be solved and why a particular one is a priority amidst other competing issues. Policy entrepreneurs bind the three MSF streams together and create the policy window which advances the policymaking process.

To understand the rationalization of the government’s actions, the concept of governmentality and its implication on governance is applied. Michel Foucault describes governmentality (from the two words, government and rationality) as the process whereby governments exercise rational and carefully considered programs meant to be undertaken by diverse agencies and entities deemed suitable for the societal good [21]. In this view, citizens are perceived as willing participants to be governed by the elite and legitimize this participation through constituted norms. Norms refer to the implicit informal ideas and social behavior that is “constructed, understood, and disseminated among groups through communication” without resistance [22]. Acceptance of government decisions without opposition conveys layers of power dynamics that function through the lenses of different political strategies. Foucault refers to such acquiescence as biopower, that functions within the realm of biopolitical management [23,24]. By biopower, Foucault emphasizes how governments exercise “power that exerts a positive influence on life, that endeavors to administer, optimize, and multiply it, subjecting it to precise controls and comprehensive regulations” [25,26]. Deductively, biopower can be stretched in the interest of the government/authority to maintain stable governance. The governance style and policy frameworks/vehicle used in Rwanda are analyzed and connected to show how they influence the policymaking process.

## 3. Results

### 3.1. Problem Stream

In Rwanda, cervical cancer ranks as a leading cause of female cancer and death. Before 2011, the age-standardized incidence rate of cervical cancer was 34.5 cases per 100,000 women, with an age-standardized mortality rate of 25.4% [12,16]. Given this high prevalence, the government prioritized cervical cancer prevention and control.

### 3.2. Policy Stream

Friends of the Global Fund Africa (a.k.a. Friends Africa) inaugurated in Kigali, Rwanda in 2007 was founded to increase awareness of Global Fund ideals to fight HIV/AIDS, tuberculosis, and malaria in Africa. As a board member of Friends Africa, first lady of Rwanda Jeanette Kagame’s involvement in Global Fund activities reflects her prioritization of health issues in Africa. In April 2009, Mrs. Kagame met with Merck executives, manufacturers of the HPV vaccine, Gardasil^®^, to negotiate access to Gardasil^®^ for Rwandan girls and women. Merck has programs enabling impoverished countries to apply for free doses of the vaccine [27]. Her efforts led Merck to send representatives to Rwanda in April 2010 to work with the Rwandan Ministry of Health and other technical working groups to develop a plan to deploy a national cervical cancer strategy [16]. The relatively stable relationship between the stakeholders created a conducive policy network environment. Within six months (October 2010), the policy network led to the formation of a National Strategic Plan for the Prevention, Control, and Management of Cervical Lesions and Cancer [16]. In the plan, primary school-aged girls would be targeted to receive the 3-dose schedule of Gardasil^®^ vaccine while women between the ages of 35 and 45 years would undergo routine screening. The rationale was that about 98% of Rwandan girls attend primary school, and women 35–45 may have already debuted sex in their lifetime [1,16]. The effective participation and buy-in of the Ministry of Education was crucial to program success as a school-based program would enable significant coverage. The working group targeted girls in primary grade six, with the expectation that most of them may not have debuted sex [16]. The working group also targeted girls not in school through a community-based strategy [16,27]. In the early stages of the HPV vaccine, many countries, especially in HICs that debated HPV vaccination policy, were conflicted on where the focus should be; whether to focus public information campaigns on the transmission of the disease or “on the fact that HPV leads to cancer and this vaccine will prevent cancer” [15]. In its planning stage, Rwanda focused its emphasis on cancer prevention. According to the Minister of Health at the time, Agnes Binagwaho, “the Ministry of Health considered the overwhelmingly positive evidence of the effectiveness of the HPV vaccine to be a call to action” [16].

### 3.3. Politics Stream

Despite the stable policy environment in the Rwandan HPV vaccination policymaking process, stakeholder resistance was not absent from the process. Typical of this is Nobila Ouedraogo and colleagues, who wrote a correspondence letter to the Lancet editor to express their dissatisfaction with the Rwandan HPV vaccination program in July 2011. Ouedraogo and colleagues expressed “serious doubts that this arrangement [referring to the Rwanda and Merck arrangement] is in the best interest of the people” [28]. The authors criticized the government for being secretive about the cost of the vaccine, choosing to eliminate cervical cancer when other vaccine-preventable diseases such as tetanus and measles needed prioritization, raised concern about the uncertainty around the effectiveness of HPV vaccines, and finally claimed issues of conflict of interest [28]. The Rwandan Minister of Health and her colleagues responded to the arguments by Ouedraogo and colleagues in correspondence to the Lancet editor (see Table 1).

Binagwaho and colleagues, in conclusion, remarked that Ouedraogo and colleagues’ perspective “reminds us of nihilistic claims against provision of antiretroviral therapy in Africa”, one “that constitutes but the latest backlash against progressive health policies by African countries” [15]. External adversaries did not resurface after Binagwaho and colleagues’ response, thus, allowing the policymaking process to maintain the stable policy network formed to formulate and implement the program. Suffice it to say, in the early days of Gardasil^®^ approval, there were concerns about the safety of the vaccine particularly in some HICs when their HPV vaccination programs were launched. Gardasil^®^ was approved by the U.S. FDA on 8 June 2006, and by 1 June 2009, approximately 25 million doses had been distributed to girls and women between the ages of 9–26 years. During this period, the U.S. Vaccine Adverse Event Reporting System (VAERS) recorded 53.8 adverse effects per 100,000 vaccine doses [29]. Some activists capitalized on the VAERS data to intensify the vaccine safety controversy [27,30]. Before the Rwandan nationwide HPV vaccination in 2011, however, research on the safety of the vaccine led vaccine programs to continue in most HICs [31,32,33,34]. The implication of the series of studies on the vaccine’s safety provided a reasonable basis for Rwanda to proceed.

### 3.4. Policy Entrepreneurs

Many external interest groups, such as Merck, Qiagen, Gavi (the Vaccine Alliance), CDC, and the International Center for AIDS Care and Treatment Programs (ICAP) at Columbia University, and critical internal actors, particularly Jeannette Kagame, played various vital roles. As the Rwandan government lacked the financial resources to fund the program on its own, Merck’s role in instrumentalizing the project with technical strategies, program development, and donating vaccines along with Gavi’s on-going support fostered the enabling conditions.

### 3.5. Policy Window

The arrangement between the government of Rwanda and Merck led it to donate two million doses of Gardasil^®^ over three years, which opened the window of opportunity. Related agreements by the Rwandan government engaged Qiagen and Gavi to ensure the flow and continuity of Rwanda’s effort to eliminate cervical cancer in the country. After the three-year arrangement with Merck concluded, Gavi agreed to cover the cost of the vaccines supplied by Merck. Moreover, Qiagen provided 250,000 HPV tests for women aged 35–45 in Rwanda as part of the cervical cancer prevention program. These arrangements constitute what Binagwaho and colleagues referred to as a “public-private community partnership for effective [program] implementation specific to the Rwandan context” [14]. In 2011, the Rwandan government dedicated 22.1% of the country’s budget (about 11.0% of the GDP) to the health sector [2]. According to Holmes, Rwanda’s attitude to foreign aid for health is a crucial indicator of success as it “fully integrated [aid] into the health system, and is only used if it addresses a need already identified by the Ministry of Health” [1] (see Figure 1).

### 3.6. Policy Network Stability

The action by policymakers to take advantage of the window of opportunity depended on the policy network’s stability. When a policy network is stable, a policy equilibrium is maintained where stakeholders are willing to negotiate on some of their inherent interests for the collective good of the network [35,36,37]. HPV policy and program proposals received less resistance, unlike in other high-income jurisdictions such as the U.S. This stability, along with such factors as memories of the 1994 genocide and its devastating socioeconomic impacts, is cautiously believed to be due to a national commitment to rebuild a broken country in unity rather than in disunity. Determining the extent to which this is a factor would require consideration of the broader politics in Rwanda including where the concerns of other interest groups can be situated given the alternatives regarding a policy problem. To answer this will require a much lengthier analysis beyond the scope of this paper. However, we attempt to address this question.

Rwanda has three ethnic groups, the Hutu (85%), Tutsi (14%), and Twa (1%). Whereas the Hutus are the majority, political power has predominantly been held by the Tutsis [9]. Before the war in 1994, the Rwandan Patriotic Front (RPF) Party had oversight of political power and was dominated by minority Tutsis. The Tutsis also own significant enterprises and businesses in the country. The power imbalance and socioeconomic inequity at the time were flashpoints for the war in 1994. In post-war Rwanda, these problems are addressed in an inclusive governance approach through thoughtful power distribution, decentralization, and ownership of government-led programs. With these structures, governments will expect little to no resistance. For example, Chemouni emphasizes the existence of virtually no political opposition to government policies, thus preventing the “emergence of alternative political ideas and projects” [9]. Two suggestions for this positioning are posited that: (1) the Rwandan Constitution limits an incumbent political party from holding more than 50% of ministerial portfolios, and (2) limits on media and civil society activities are normalized [9,38,39,40].

Some commentators questioned the near absence of opposition voices in the Rwandan political and policymaking process and criticized the incumbent government led by president Paul Kagame as running a one-party state where opposition to social policies and programs is not tolerated [9,41,42,43]. To understand this criticism, Foucault’s analysis of government of the living using techniques, tools, technologies, procedures, and processes to direct behavior is instructive as these mechanisms predispose the population to discipline and social order through what he referred to as biopower and biopolitics. In Rwanda, we can abstract that biopower and biopolitics symbiotically are central to the government strategy of achieving policy and program goals. Ideals such as Ubudehe and Imihigo, which are locally self-managed strategies, can be thought of as underlying biopower and biopolitics at play. According to the Rwandan online portal, https://rwandapedia.rw/hgs/ubudehe/overview (assessed on 22 April 2023), Ubudehe is a social welfare term that “refers to the long-standing Rwandan practice and culture of collective action and mutual support to solve problems within a community”. Ubudehe by its design decentralizes government power and allocates resources to meet the social needs of the people [44,45,46]. For example, key informant (R001) who represents a women’s group in Kigali indicated that “Every Rwandan is entitled to health insurance. The Government pays insurance for the poor identified by Ubudehe (Levels according to socio-economic status)”. (R001, 9 December 2021). Similarly, Imihigo signifies pledges and is a performance evaluation framework that decentralizes responsibilities of government-initiated projects, holds local and central leaders at all levels responsible for ensuring predefined project targets are met, and promotes accountability and ownership of same [16,20].

Assessing the posture of network stability pushes to the fore the development aid Rwanda receives. In Aid and Authoritarian Africa, Bird asserts that in Africa, aid can become a tool to accentuate power in different forms. For example, he posits that while the Kagame government is lauded in development areas such as health and education, “opposition voices and dissent are regularly suppressed” [47]. Kagame, a former military leader, has been likened to the Italian diplomat and politician Machiavelli and his political ideals in his famous book The Prince [48,49,50]. According to Reese, Kagame’s political leadership style “inspires love, fear, and a unique paternalism” among Rwandans [48]. His Machiavellian leadership style has led the country to be perceived as successful with special attention in Africa (ibid). Russel has called Kagame’s leadership a “benevolent dictatorship” that offsets negative government outlook and militant invasion in the Democratic Republic of the Congo for positive outcomes such as security and stability for its citizens [51]. The leadership model for an individual with military background stepping into a democratic space hinges on “two attitudinal changes—democratized decision-making and adapted political goals” [52]. Waldorf posits that while the Kagame regime adapted its political goals of rebuilding the nation to appease political opponents, the government did not democratize the model for decision-making. The undemocratization of decision-making at once becomes a tool and technique that beguiles fear on one end and obeisance on the other, thus, maintaining a powerful tool that can implicitly or explicitly quieten policy and political adversaries, “re-educate the populations, deliver public goods, and attract donors and investors” [52]. Presumably, policy network stability in Rwandan policymaking processes presents a distinct view of political power and dominance, a bold leadership style, and a culture of policy acceptance rather than engagement. The approach reflects nuances that drive policy success and wades off policy failure from the onset. Kagame’s perceived protectionist style of policymaking can build barriers to policy alternatives as the process blocks valuable ideas that may never be shared due to the stable policy network environment which the protective policy network fosters.

### 3.7. Local Policy Accountability Frameworks

Rwanda has a track record of achieving very high (over 90%) childhood vaccination coverage in children under five years for diseases such as diphtheria, Haemophilus influenza type B, pertussis, measles, polio, tetanus, and tuberculosis. This success has positioned Rwanda as attractive to donor agencies like Gavi. According to Bao and colleagues, post-war Rwanda has consistently leveraged “strong relationships with development partners and cross-over effects from global health initiatives, particularly in developing capacity for supply chain and cold chain management” regarding vaccination programs [20]. Agnes Binagwaho remarked in an interview with Lancet that when it comes to vaccination, support from international development organizations is a significant gain [1]. The country effectively organizes and integrates external support received and couples it with internal resources, norms, and systems, such as Imihigo. Imihigo has been instrumental in Rwanda’s universal childhood vaccination coverage and was an essential tool in the HPV vaccination program’s success. Markers like Imihigo reflect well on making it much easier to request support where needed. It is in this light that Merck positioned its interest (either financial or social) to become an active player as a policy entrepreneur in the Rwandan HPV vaccination program. According to key informant (R001), HPV vaccination in Rwanda thrives on “free vaccinations and advocacy including our first Lady”. She further noted that the “national and private televisions plus newspapers” influence vaccination uptake in Rwanda (R001, 9 December 2021). Mrs. Kagame’s role as a policy entrepreneur advocating for cervical cancer elimination in Rwanda and her engagement with Merck and other stakeholders for support is consistent with the country’s outlook on aid and capacity building to improve health that triggered an alignment of the policy, problem, and politics streams.

## 4. Discussion

The study elucidates how health prioritization stimulates political will and mobilizes resources for policy action and sought to clarify the HPV vaccination policymaking process in Rwanda while distilling lessons that can inform other LMICs.

### 4.1. Planning and Prioritization

Vaccine-targeted HPV types in Rwanda are reportedly decreasing since the introduction of the nationwide HPV vaccine program [53]. Estimation of age-standardized incidence rate for cervical cancer in Rwanda was 28.2 per 1,000,000 women in 2020, even though this is higher than the global estimate of 13.3 per 100,000 women [54,55]. Cervical cancer is still prevalent in Rwanda, however, with the decreasing trend in vaccine-targeted HPV types of infection and age-standardized incidence rates for cervical cancer, it is evident that the nationwide HPV vaccination program is yielding dividends in the overall health of the population.

The Rwanda genocide in 1994 may have created debilitating effects on the country’s healthcare system. However, understanding the health challenges faced and leveraging opportunities to prioritize population health, set it up for success. The Rwandan HPV vaccination policymaking process with the active role of elite actors such as Mrs. Kagame, effective harnessing and management of policy actors’ expertise, utilization of local policy frameworks, and governance facilitated success amidst challenges. Rwanda’s HPV vaccination program and the policymaking process reveals resilience considering the country’s past war history and post-war improvements made in the healthcare system. In Rwanda, constraining factors challenged the nationwide HPV vaccination program, however, these were countered through adequate planning and resource use (see Figure 2).

Prioritizing health and obtaining required political will from key stakeholders such as government for action does not guarantee policy success when planning is ineffective, and execution cannot be adequately advanced with the ability to address challenges.

### 4.2. State-Non-State Relationships

Policy traverses a continuum (success to failure) and the factors defining policy outcomes pivot around the actors and their position on the policy. Government is an important determinant of a policy success or failure due to its access to state and some non-state resources. In Kingdon’s view, policy entrepreneurs may seek to place their resources where they can advance future policies to which they subscribe. Merck as a policy entrepreneur, played a critical role in Rwanda’s HPV vaccination program as the relationship defined how the State and the private sector can create shared value through the utility of resource mobilization and effective management. In an analysis of the politics of the HPV vaccination policymaking process in the United States, for example, Abiola and colleagues found that “effective policy entrepreneurship played a critical role in determining policy outcomes” [56]. In an assessment of pharmaceutical companies and their involvement in vaccination policymaking in six states from 2006–2008, for example, Merck was found to participate in providing “scientific information about Gardasil^®^ or [provided] potential policy strategies” [57]. While Merck was found to be a dominant player in the American HPV policymaking process, some backlash against inappropriate financial inducement emerged along with political misjudgment by legislators influenced policy options for nationwide HPV vaccination [56]. During a congressional hearing in the U.S. House of Representatives, Michelle Bachmann, accused Texas Governor Rick Perry of conflict of interest and misconduct for ordering sixth grade girls to receive HPV vaccination because of his financial and political relationship with Merck. Such influence on American HPV vaccination policymaking processes may have given Ouedraogo and colleagues cause to doubt the arrangements between Merck and the Rwandan policy process, particularly the Minister of Health, Agnes Binagwaho, and Merck’s prior relationship as board members of Gavi. In the case of Rwanda, the response from Binagwaho and colleagues ended the emergence of future adversaries of the Rwandan HPV vaccination program. Their response was pivotal in paving the way for the HPV vaccination program in Rwanda as the country pioneered their agenda for cervical cancer prevention in Africa.

The arrangement between Rwanda and Merck meant Merck would discount its profit for future returns by donating two million vaccine doses through the Merck Medical Outreach Program [58]. Rwanda capitalized on this opportunity, utilizing its local frameworks, such as Ubudehe and Imihigo, and bringing all relevant stakeholders together for policy action. The alignment of policy actors generally will include the government, society, and corporations. The intersection of these three realms does not always produce policy equity, however, as in most cases, government and corporations tend to bind tightly together in the policymaking process. Understandably, the primary objective of government is to satisfy their fiduciary duties as a foremost responsibility to the population, and for the vaccine manufacturer, to its shareholders. For this reasons, as Perkins asserts, “[vaccine manufacturers] must make decisions based on profit” [59]. According to Ledley and colleagues, understanding this fundamental base of the pharmaceutical companies as a profit making entity is “essential to formulating evidence-based policies to reduce [medicine] costs while maintaining the industry’s ability to innovate and provide essential medicines” [60]. It is in the interest of Merck to make the vaccine donation to Rwanda and other LMICs because the action not only reflects corporate social responsibility, but also a marketing tool that provides social license to operate and channel to sell medicines to LMICs at reasonable cost and terms.

In 2011, Gavi provided opportunities for countries to apply for funding for HPV vaccines [61] if they met the following conditions (i) at least US $1580 in Gross National Income per capita and (ii) achieved at least 70% coverage for Diphtheria-Tetanus-Pertussis third dose (DTP3) and similar vaccines, while demonstrating the capacity to deliver multiple dose vaccines to children from the ages of 9–13 years at 50% coverage [62]. Once Gavi has earmarked a country for assistance, support is initially provided to gain implementation knowledge by conducting demonstration vaccination [62]. The demonstration offers an opportunity to streamline challenges prior to proceeding to national vaccination by leveraging the knowledge gained during the demonstration stage. This is usually for a period of two years at most, if the first-year demonstration did not provide enough evidence that replication at the national level will be successful. Besides the support to purchase the vaccine, Gavi also provides substantial funding to offset about 80% operational cost of the vaccine introduction at the discretion of the countries (ibid). While the market price for the HPV vaccine is around $100, through Gavi’s assistance, with support from the WHO, vaccine manufacturer (Merck), the World Bank and the Bill and Melinda Gates Foundation, (BMGF), the HPV vaccine is made available to LMICs at a price of $4.50 per dose [63]. Whereas this is a significant reduction in price, it could stretch healthcare expenditures for some LMIC countries, thus, potentially making the decision to purchase a challenging one for resource-constrained nations.

The driving force of public-private- partnership is the harnessing of efforts and creation of shared value, mainly one that leads to the creation of social value culminating in improved health, especially in resource-constrained nations [64]. In the public-private- partnership agreement, Rwanda leveraged Gavi to assume a payment arrangement with Merck as a continuity package for program progression. With this arrangement, Merck eventually offset the lost margin on the two million doses donated while enjoying an extended financial return over the program’s lifespan. Ruckert and Labonté noted that partnership between the public and private sector includes “neoliberal management of individuals and populations, allowing private interests to become embedded within the public sphere and to influence global and national health policy making” [65]. The authors cited the RotaTeq Nicaragua Partnership between the Nicaraguan Ministry of Health and Merck, local hospitals, and a Technical Advisory Group to successfully implement a rotavirus vaccination campaign in Nicaragua [65]. Merck’s experience in public-private partnership in vaccine program development was another crucial factor in Rwanda’s HPV vaccination roll-out. While the role of Merck was criticized as not in the best interest of the Rwandan people [28], the success of the program suggests that input from vaccine manufacturer in the policymaking process shaped the policy outcome.

The WHO and Gavi have increasingly called for incorporation of HPV vaccination in national immunization programs considering the vaccine’s efficacy, safety profile, and cost effectiveness. The WHO has indicated that this must be done within consideration of national public health prioritization, adequate financial sustainability, and cost effectiveness of the vaccination program [59,60,61]. While countries such as Rwanda have made inroads in incorporating HPV vaccination into their national immunization programs, many more still struggle to attain policy convergence due to lack of political will, inertia, social, and economic considerations.

This study attempts to enhance our understanding of the actions of actors through the lens of Kingdon’s multiple stream framework, delineating the governance structure within Foucault’s concept of governmentality and its policymaking implications. To the best of our knowledge, this is the first study to clarify the Rwanda nationwide HPV vaccination policymaking process from this perspective.

### 4.3. Limitations

Although our methodological approach of triangulation across government and literature sources in which HPV vaccine program designers shared their insights along with key informant interviews offers a sound approach, additional key informant interviews could have further clarified the Rwandan case. A scoping review on the HPV vaccination program in Rwanda would offer an opportunity for further studies.

## 5. Conclusions

Rwanda presents a valuable case study of Kingdon’s multiple streams model clarifying how governmental priority setting (policy stream) for cervical cancer prevention (problem stream) along with public and private incentives and policy entrepreneurship (politics stream) aligned to foster program implementation. Rwanda’s track record of successful vaccination programs enabled by a culture of local accountability fostered by Imihigo were important factors that led to public and private sector incentives. While public-private partnerships can be contentious, when well-managed, they can create symbiotic value streams that all stakeholders can leverage for their long-term interest. Rwanda’s community pledge, Imihigo, is successful because people are held responsible for government-initiated programs such as nationwide vaccination programs. At the same time, the Kagame leadership style also influences people’s behavior and becomes a policy instrument that shapes the country’s policymaking process. Political will on the part of the Rwandan government articulated through the policy entrepreneurship of Jeannette Kagame and the policy network established with Merck and Gavi were foundational to the health policy outcomes. Even though some critics argue that President Kagame’s government is intolerant of policy resisters, his leadership has, nevertheless, assembled the political tools, policymaking elements, practices, and thinking of policymakers to address public health issues such as vaccine-preventable diseases effectively. The Rwandan HPV vaccination program is a unique case in Africa whose replication in other low-resource settings without similar policymaking scenarios, such as communal norms of policy responsibility (e.g., Imihigo), an effective immunization track record, and skills to effectively leverage aid, will be complicated; however, not impossible. While Rwanda presents several policy lessons, it is imperative that low-resource countries attempting to implement a nationwide HPV vaccination program concentrate on their unique strengths in policy design, strategic program development, plans for resource mobilization, and finally, design a policy evaluation tool that serves to measure markers of success.

## Figures and Tables

**Figure 1 ijerph-20-06998-f001:**
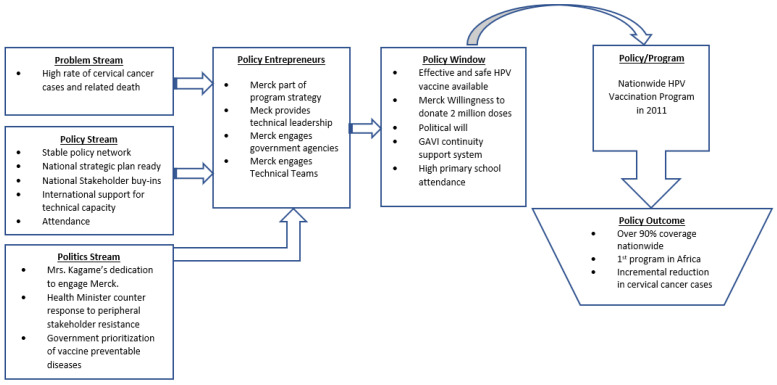
Flowchart of Kingdon’s MSF framework used to clarify the HPV vaccination policymaking process in Rwanda.

**Figure 2 ijerph-20-06998-f002:**
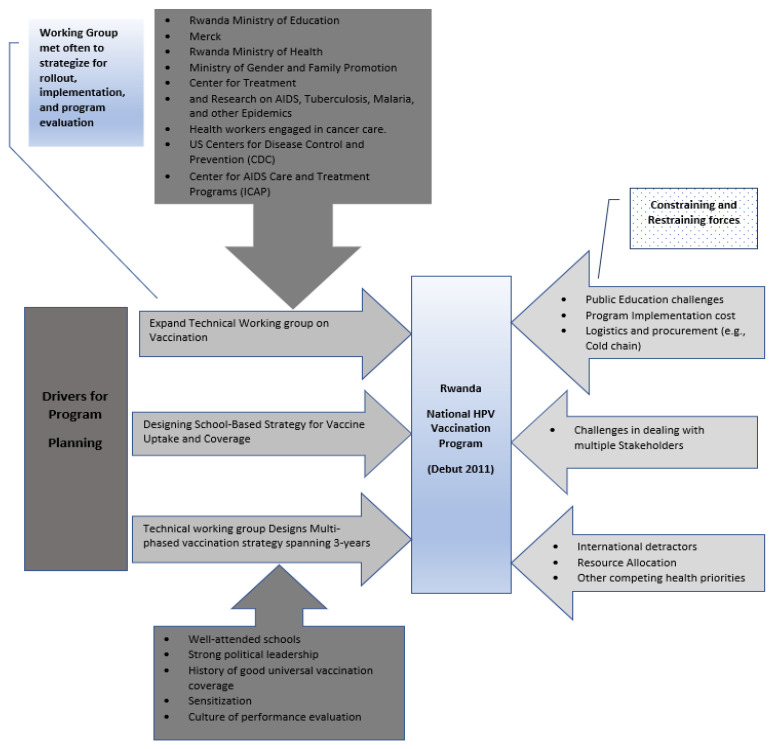
Schematic diagram of drivers of the Rwanda nationwide HPV vaccination program.

**Table 1 ijerph-20-06998-t001:** Peripheral stakeholder resistance and maneuverability.

Argument ^a^	Counterargument ^b^
We have serious doubts that this arrangement Merck providing HPV vaccines to Rwanda] is in the best interest of the people.	Are the 330,000 Rwandan girls who will be vaccinated against a highly prevalent, oncogenic virus for free during the first phase of this programme not regarded as “the people”?
Although the burden of cervical cancer in low-income and middle-income countries is substantial (3 · 8 million disability-adjusted life-years [DALYs]), it ranks well behind that of other vaccine-preventable diseases such as tetanus (8 · 3 million DALYs) and measles (23 million DALYs).	For the diseases cited (measles and tetanus), Rwanda has 95% and 96·8% vaccination coverage rates, respectively.
The effectiveness of the HPV vaccine against cervical cancer is still unknown.	Many studies say otherwise.
To remain cost-effective in GAVI-eligible countries, the costs for a vaccinated individual should not exceed US$10 for the three doses.	The initial price of the pneumococcal vaccine provides a helpful lesson, and Merck announced a two-thirds reduction in the price of Gardasil for GAVI-eligible countries (to US$5 per dose).
Representatives of vaccine manufacturers and the Rwandan Minister of Health are on the GAVI Board—an obvious conflict of interest.	Merck representatives are non-voting GAVI observers, and GAVI’s website clearly shows Rwanda’s board membership terminating on 31 December 2011. GAVI will have no role in the HPV vaccine program before 2014.

^a^ [28] ^b^ [15]; GAVI (aka Gavi, the Vaccine Alliance; formally known as the Global Alliance for Vaccines and Immunization) is a public-private global health partnership that seek to improve access to vaccines in resource constrained countries.

## Data Availability

The data are available from the corresponding author upon reasonable request.

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
