# Peer review of "Understanding HPV Vaccination Policymaking in Rwanda: A Case of Health Prioritization and Public-Private-Partnership in a Low-Resource Setting"

_ijerph, 2023, doi:10.3390/ijerph20216998_

Round 1
Reviewer 1 Report
Comments and Suggestions for Authors
This review of HPV immunization in Rwanda is very interesting since it is a triumph of implementation and a model for other countries.
1. Not much detail about the key informant interviews.
2. Too much detail about the politics. Much of it is not relevant.
3. Need more information about outcomes: decreased need for screening with Pap tests, rates of warts, HSIL rates and what was the effect on male HPV diseases?
4. GAVI: not defined; not clear what its role was here.
Reviewer 2 Report
Comments and Suggestions for Authors
I have thoroughly reviewed the manuscript titled "Understanding HPV Vaccination Policymaking in Rwanda: A Case of Health Prioritization and Public-Private-Partnership in a Low-Resource Setting," which aims to elucidate the process of formulating HPV vaccination policies in Rwanda from the perspective of Kingdon's multiple stream framework and Foucault's concept of governmentality. While I appreciate the manuscript's premise, I must express concerns about its execution.
The authors have succeeded in developing a solid introduction that appropriately contextualizes the issue at hand. However, significant structural issues are identified throughout the text:
1. Abstract: The abstract includes information that is not authored by the researchers and, therefore, should not be included in this section. An example is the assertion that Rwanda was the first African nation to initiate a national HPV vaccination program in 2011, with specific statistics regarding the reduction in cervical cancer incidence and mortality rates. It is essential that the abstract contains only information generated within the study itself, without references to external sources.
2. Method: Methodology is essential to impart scientific rigor and robustness to the work. The authors mention the use of qualitative methods, including literature review and interviews with key informants. However, crucial details are missing. It is imperative that the authors provide information about the literature review, including who conducted it and how it was conducted. Similarly, it is important to clarify who the key informants are, how they were selected, and how the interviews were conducted. The absence of these details compromises the credibility of the results.
3. Results: The presented results lose their validity due to the lack of a robust methodological description. Readers need to understand the origin of the results to assess their reliability and relevance. Therefore, it is essential for the authors to provide detailed information about data collection and analysis.
4. Discussion: The discussion section is brief and, at times, conflates with the results. Moreover, the inclusion of a figure in the discussion requires a clear logical explanation. The discussion section should be separated from the results and provide an in-depth analysis of the results in the context of relevant literature.
Reviewer 3 Report
Comments and Suggestions for Authors
The manuscript presented for review discusses the up-to-date topic of gynecology and reproductive medicine. Although cervical cancer screening programs and HPV vaccination is well developed, cervical cancer incidence is increasing in many countries worldwide. Since 2006 when the approval of the first HPV vaccine occurred by June 2020, 55% of the WHO Member States reported implementation of partial or national HPV vaccination..
Thus, any efforts to improve the public knowledge of cervical cancer and HPV vaccination remain valuable. Therefore, the manuscript has significant value for modern medical practice and could be published after some requested revisions are done.
Please find below my comments:
1. Section 2 2. Theoretical Lens should be moved to the methods part.
2. The methods part is incomplete. It should include the following sections:
Study subjects, exclusion and inclusion criteria, and study instrument (survey) should be presented in detail, participants’ recruitment procedure (if applicable).
3. Results are interesting and supported by clear tables anf figures .
4. In the discussion part is narrow and difficult to comprehend in the current version. It requires restructuring. The following outline is suggested:
Outline for the Discussion part
1.1 Rationale of the study (why it was done)
1.1.1 Main findings of the study
1.1.2 What makes your study unique
1.1.3 What it adds to what we already know
1.2 Study subjects
1.3 Subject of the discussion
Comparison of your results with previous studies in the field. Agreement and disagreement with the studies compared
1.4 Study strengths and limitations
1.5 Clinical implication
Round 2
Reviewer 2 Report
Comments and Suggestions for Authors
The authors did a good review of the material.